# Feasibility of Virtual Shopping Budget-Management Training on Executive Functions in Healthy Young Adults: A Pilot Study

**DOI:** 10.3390/brainsci13111573

**Published:** 2023-11-09

**Authors:** Si-An Lee, Ji-Yea Kim, Jin-Hyuck Park

**Affiliations:** 1Department of ICT Convergence, The Graduate School, Soonchunhyang University, Asan 31538, Republic of Korea; iop5213@naver.com (S.-A.L.); kimkim2@sch.ac.kr (J.-Y.K.); 2Department of Occupational Therapy, College of Medical Science, Soonchunhyang University, Asan 31538, Republic of Korea

**Keywords:** executive functions, virtual reality, IADL, shopping program, budget management, fNIRS

## Abstract

To date, budget management in virtual shopping training has not been given much importance. The main objective of this study was to investigate the effects of virtual shopping budget-management training on executive functions and brain activation. Sixteen participants were randomly assigned to the experimental group that received virtual shopping budget-management training or the waitlist control group for a total of 16 sessions. To examine the effects of virtual shopping budget-management training on brain activation, HbO_2_ was measured in the prefrontal cortex via functional near-infrared spectroscopy (fNIRS) during the Trail Making Test Part B (TMT-B) and Stroop test. Mann–Whitney and Wilcoxon signed-rank tests were used to compare outcomes between and within the two groups. The virtual shopping budget-management training showed no significant difference in all outcomes between both groups (*p* > 0.05). No significant differences were observed in HbO_2_ levels during both TMT-B (*p* > 0.05) and the Stroop test (*p* > 0.05). However, in the pre-post comparisons, there was a significant difference in the TMT-B (*p* < 0.05) and Stroop test (*p* < 0.05) in the experimental group. In this study, although we did not find a distinct advantage in training, it confirmed its potential for clinical benefits in healthy young adults through training.

## 1. Introduction

Executive functions consist of high-level cognitive processes that enable individuals to function independently to facilitate new behavioral patterns and optimize approaches to unfamiliar environments, comprising sub-elements such as inhibitory control, working memory, cognitive flexibility, reasoning, problem-solving, and planning [1,2]. Additionally, instrumental activities of daily living (IADL) encompass a range of activities crucial for independent living [3]. The absence of appropriate executive control may result in a higher likelihood of encountering difficulties in performing IADL effectively [4]. Furthermore, because executive function impairment can have a significant impact on self-regulation [5], it can lead to problems with IADL that are more complex and require a higher level of personal autonomy for independent living, such as shopping [6]. Symptoms of executive dysfunction include slow initiation, inability to suppress automatic responses, poor problem-solving ability, and use of inappropriate rules [7], which can reduce independence in IADL [8]. Therefore, improving executive functions for independent IADL can be the goal of rehabilitation.

Conventional cognitive interventions aimed at enhancing executive functions primarily focus on improving sub-components of executive functions through demanding cognitive training with tabletop activities and computerized cognitive training programs, which limits the integration of overall executive functions [9]. Moreover, since their contents are far from real-world, it is difficult for their effects to be transferred to the subject’s daily life [10]. Therefore, cognitive training for executive functions should consider ecological validity, rather than focusing on sub-elements of executive functions [11].

Virtual reality (VR) is being widely applied to ensure ecologically validated cognitive training. It has the advantage of providing cognitive training in a safe environment, personalizing the situations, tasks, and difficulty of training, and providing feedback by tracking and analyzing the participant’s functional improvement [12]. Indeed, the ecological validity of VR-based cognitive training has been confirmed through a meta-analysis study [13], supporting the feasibility of VR for cognitive training.

Recently, virtual shopping training has been developed to assess and treat executive function impairment [14,15,16]. This training consists of having the subject remember shopping items, navigate the inside of the supermarket, and find and purchase them on the shopping list within their budget [14]. Spending and managing money within a budget are complex IADLs that heavily depend on intact executive functions [16]. However, in traditional VR-based shopping training, budget management typically involves the straightforward selection of products within a predetermined budget, rather than the creation of an optimal purchasing strategy. This approach might not fully engage executive functions [17]. Accordingly, virtual shopping budget-management training might yield superior outcomes through the optimization of executive functions compared to traditional VR-based shopping training.

On the other hand, previous studies have relied on paper-and-pencil-based executive function assessments to evaluate the effect of training [15,18]. These assessments, however, have limitations in objectively substantiating the basis for training due to their inability to observe clear changes in brain function [15]. Clear changes in brain function can be confirmed by observing brain areas related to executive functions through brain imaging techniques. One such brain area associated with executive functions is the dorsolateral prefrontal cortex (dlPFC). The dlPFC is conventionally acknowledged to play a crucial role in executive functions, which are high-level cognitive functions that support flexible goal-directed behaviors [19,20]. Brain imaging techniques such as functional magnetic resonance imaging (fMRI), electroencephalography (EEG), and functional near-infrared spectroscopy (fNIRS) have been used to measure changes in brain function, which could be evidence of cognitive training efficacy. Of brain imaging techniques, fNIRS non-invasively measures brain activity using changes in light absorption in the brain [21,22]. fNIRS has the advantages of portability, movement tolerability, and safety of use compared to other neuroimaging modalities [23].

Therefore, the primary objective of this study was to investigate the effects of virtual shopping budget-management training on executive functions and brain responses in healthy young adults as a pilot study before applying it to patients with executive dysfunction. This study hypothesized that virtual shopping budget-management training could be effective in improving executive functions and inducing changes in brain responses in healthy young adults.

## 2. Materials and Methods

### 2.1. Design

This study was a pilot study, and all participants were randomly assigned, with 8 participants in each, to either the experimental group or the wait-list control group using the Python computer language. The intervention consisted of 16 training sessions conducted twice a week over 8 weeks.

### 2.2. Participants

Sixteen healthy young adults participated in this study. Participants included in this study were individuals who had no visual or auditory impairments, were not undergoing behavioral interventions for cognitive enhancement, demonstrated comprehension of verbal instructions, and had no limitations in basic activities of daily living. Exclusion criteria comprised individuals who (1) had been diagnosed with psychiatric disorders, (2) had been diagnosed with neurological conditions, and (3) expressed an aversion to computer usage.

### 2.3. Intervention

The virtual shopping budget-management training was conducted twice a week for a total of 16 sessions, each lasting 15 min, over 8 weeks. The experimental group received the virtual shopping budget-management training program created with Scratch (MIT Media Lab, Cambridge, MA, USA). This program is an extension of conventional virtual shopping training programs, featuring added elements of budget-management training. The existing program was designed to simulate shopping in a supermarket. Participants navigated within a virtual supermarket, purchasing items from a shopping list. They had the freedom to navigate the virtual supermarket, selecting and purchasing items by touch. To make a payment, they were required to locate a cash desk, which then prompted the payment screen to appear. Following the payment, a statistical screen with feedback on their performance appeared [14]. In another program, participants performed the Adapted Four-Item Shopping Task. This task involved four shopping lists, with items positioned in two different aisles. Participants were provided with the lists, eliminating the need to memorize the items [24].

The virtual shopping budget-management training program introduced an added complexity by incorporating “scores” as variables assigned to product prices—lower prices resulted in lower scores, whereas higher prices yielded higher scores. Participants were asked to maximize their scores within a predetermined budget while retaining knowledge of their scores. This program places a particular emphasis on budget management within executive function training. It offers four levels of difficulty, challenging even healthy young adults, and utilizes various budget allocations to reduce learning effects (Figure 1).

### 2.4. Measurement

#### 2.4.1. Primary Outcome

Participants’ executive functions were assessed using the Trail Making Test Part B (TMT-B), the Stroop test. TMT-B requires participants to connect numbers and letters alternately in a consecutive sequence [25]. This assessment is related to executive function and evaluates cognitive flexibility [26]. The time it took for participants to complete the TMT-B within 300 s was analyzed.

The Stroop test typically comprises 2–4 performance tasks. In this study, participants engaged in the “interference color naming” task, where they named the color of words written in ink different from the word’s meaning [27]. This assessment evaluates cognitive flexibility and working memory [28,29,30]. There were four colors: red, yellow, blue, and black. Participants were requested to read a total of 112 words as quickly as possible. The Stroop test results were analyzed in terms of the time taken to complete the task within a maximum of 300 s.

#### 2.4.2. Secondary Outcome

Also, participants’ activation in the dlPFC area was assessed through fNIRS (Octamon, Artinis, The Netherlands), measuring hemodynamic responses within the dlPFC area. The fNIRS comprised a total of 8 channels, utilizing 760 nm and 850 nm infrared to detect changes in the cortical concentration of HbO_2_ and HHb [31,32]. Participants wore this device while performing the TMT-B and Stroop test. Hemodynamic response measurements continued until the completion of all assessments. Resting periods of indefinite duration were provided between each assessment until stability in activity was achieved. In this study, the mean values of HbO_2_ in the left and right dlPFC areas were utilized to measure hemodynamic responses. All data were sampled with a frequency of 10 Hz with Oxysoft version 3.0.52.

### 2.5. Statistical Analysis

All outcomes were analyzed using the SPSS 22.0 version (SPSS Inc., Chicago, IL, USA). To assess the normality of the sample, the Chi-square test was employed. The Mann–Whitney U test was used to analyze the pre- and post-intervention results between groups, while the Wilcoxon signed rank test was used for pre- and post-intervention analysis within the group. A statistical significance was set as *p* < 0.05.

## 3. Results

The flow of the study is illustrated in Figure 2. A total of 16 participants were randomly assigned, forming the experimental group with 8 participants and the control group with 8 participants. Following the pre-test, one participant in the control group dropped out. The results were analyzed using data from 8 participants in the experimental group and 7 participants in the control group.

Table 1 presents demographic characteristics, showing no significant differences between the two groups (*p*’s > 0.05) (Table 1).

### 3.1. Primary Outcome

In the pre-intervention, there were no significant differences between groups in both TMT-B (U = 20.000, *p* > 0.05) and the Stroop test (U = 24.000, *p* > 0.05) (Table 2). In the post-intervention, there were no significant differences in TMT-B (U = 24.000, *p* > 0.05) and the Stroop test (U = 24.000, *p* > 0.05) (Table 3). However, when comparing changes within each group, significant differences were found in the experimental group for both the TMT-B (Z = −2.38, *p* < 0.05) and Stroop test (Z = −2.38, *p* < 0.05). In the control group, a significant difference was observed in the Stroop test (Z = −2.197, *p* < 0.05) (Table 4).

### 3.2. Secondary Outcome

There were no significant differences between and within groups for HbO_2_ for the TMT-B (*p*’s > 0.05) and Stroop test (*p*’s > 0.05) (Table 2, Table 3 and Table 4).

## 4. Discussion

This study was designed to investigate the effects of virtual reality shopping budget-management training on executive functions and prefrontal brain activity. To achieve this, the experimental group participated in 16 training sessions, and significant differences in executive functions were observed after the 16 sessions of training. These results are consistent with previous research findings, demonstrating the effectiveness of virtual shopping training in enhancing executive functions [15,16].

Executive functions can commonly be categorized into three components: updating and monitoring of information (‘updating’), inhibition of prepotent impulses (‘inhibition’), and mental set shifting (‘shifting’) [33,34]. Specifically, the ability to update information about prices, inhibit the impulse to buy, and think flexibly to select the most appropriate items within a budget is crucial for reasonable consumption [35], indicating the importance of executive functions in shopping contexts. Previous research has revealed that individuals with executive function deficits, such as those with MCI or stroke, take longer to shop, exceed their budgets more frequently, and require more assistance compared to healthy individuals [14,16]. This supports the notion that shopping could serve as an effective training method for enhancing executive functions.

One significant difference between the existing shopping training and the current study is the emphasis on budget management within the shopping tasks. Existing shopping training was relatively straightforward, primarily involving the task of selecting items within a budget constraint. This simplicity facilitated the implementation of a strategy focused on purchasing the cheapest items, which might not have effectively enhanced executive functions. In contrast, the virtual shopping budget-management training introduced variability in prices across different brands, with higher prices yielding more points. To achieve higher scores within the given budget, participants had to engage in more complex mental calculations of adding and subtracting items. Consequently, this training presented a more intricate task compared to existing ones, requiring efforts to choose more suitable items within the budget. As the task complexity increased, this training closely resembled the actual shopping environment, encouraging participants to exhibit more rational consumption behavior.

Another difference is our attempt to investigate the effects of the training along with fNIRS, focusing on activation in the PFC. Neuronal activation is crucial in neurorehabilitation, as it can lead to structural and functional changes in the brain through neuroplasticity [36,37]. In this study, we measured HbO_2_ levels in the dlPFC, which is responsible for executive functions [19], and compared pre- and post-intervention differences. Although statistically significant results did not emerge, this effort is significant as the first attempt to demonstrate that virtual shopping budget-management training induces brain activation.

In this study, we confirmed improved executive functions when conducting virtual shopping budget-management training in healthy young adults. This suggests that virtual shopping budget-management training could be effective in enhancing executive functions not only in clinical groups such as MCI or stroke patients but also in healthy individuals. By focusing on budget management in shopping training and by adjusting the difficulty level by dividing it into four stages, we can provide suitable training for a diverse range of participants, including healthy adults. However, there was no difference between the experimental and control groups. This could be attributed to the assessment tool being too easy to evaluate healthy young adults, potentially resulting in a ceiling effect [38,39]. Furthermore, we conducted 16 sessions of training, which might not have been a sufficient training period to examine changes in executive functions in healthy young adults. In a previous study, it has been demonstrated that studies with more than 20 sessions of training have resulted in significant changes [40,41], whereas those with fewer training sessions have not yielded significant results [42,43,44].

This study emphasizes the importance of advanced measurement techniques, such as neuroimaging, in evaluating cognitive training. Beyond statistical significance, it is essential to measure objective evidence, such as neural efficiency, to confirm intervention effects, surpassing the limitations of traditional assessments influenced by external factors like mood states [45,46]. Additionally, despite the small sample size and non-significant results in primary analyses, our research can contribute to the development of effective virtual reality education programs for shopping budget management in the future. Hence, our study can serve as a foundational step towards creating practical virtual reality education for budget management during shopping. Furthermore, these findings can also benefit rehabilitation and elderly care programs. Training in a virtual environment enhances cognitive abilities and daily functioning, with potential for broader societal impact [47].

This study has several limitations. Firstly, the training was conducted on healthy young adults, despite the aim of enhancing executive functions. Nevertheless, the observed improvement in executive functions in this population suggests the potential applicability of virtual shopping budget-management training to various groups. Secondly, the limited sample size in this study hinders the generalization of the research findings. Thirdly, while improvements in executive functions were observed through paper-and-pencil assessments, there were no significant results in HbO_2_ levels in the prefrontal cortex, which contrasts with previous research findings [48]. Individual differences in transfer outcomes are well-documented and it can be problematic, particularly with relatively small sample sizes typically gathered in brain imaging studies [49,50]. Fourthly, despite the presence of various sub-elements in executive functions, this study only measured some aspects using the TMT-B and Stroop test. Future research should utilize a variety of executive function assessment tools for additional analysis, particularly considering a larger sample with executive function impairments.

## 5. Conclusions

In conclusion, 8 weeks of virtual shopping budget-management training does not result in improvements in executive functions or changes in brain activation when compared to the control group in healthy young adults. However, within-group comparisons revealed improvements in executive functions. The findings of this study demonstrate the potential utility of virtual shopping budget-management training in enhancing executive functions in healthy young adults. Future research should include a larger sample size and a longer intervention, particularly targeting individuals with executive function impairments.

## Figures and Tables

**Figure 1 brainsci-13-01573-f001:**
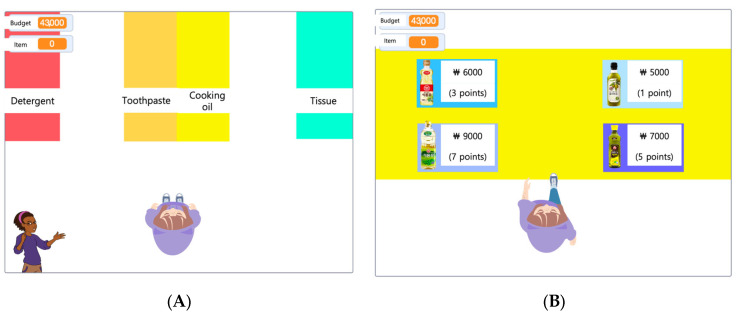
The virtual shopping budget-management training program. (**A**) Full view of the aisles. (**B**) A view of some of the shelves.

**Figure 2 brainsci-13-01573-f002:**
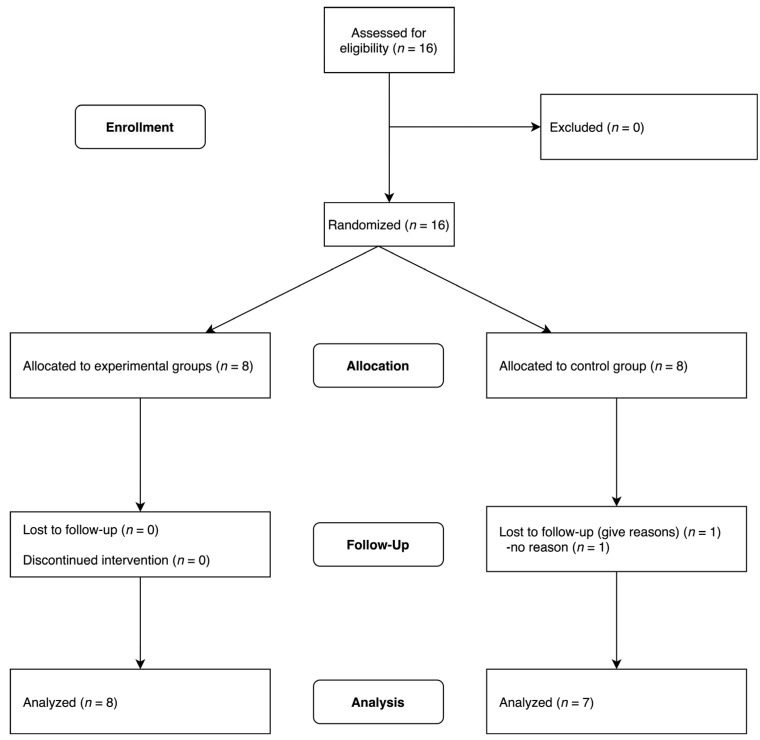
Flowchart of the study.

**Table 1 brainsci-13-01573-t001:** Demographic characteristics of the groups.

Demographic Characteristics	Experimental Group (*n* = 8)	Control Group (*n* = 8)	$\chi^{2}$
Sex: Male	0 (0 %)	3 (42.8 %)	0.055
Sex: Female	8 (100 %)	4 (57.1 %)	
Age (years)	20.38 ± 0.92	21 ± 1.60	0.771

**Table 2 brainsci-13-01573-t002:** Comparison of executive functions between both groups (pre-intervention).

	Experimental Group (*n* = 8)	Control Group (*n* = 7)	U	*p*
TMT-B (s)	19.01 ± 5.25	17.79 ± 6.75	20.000	0.397
Stroop Test (s)	87.86 ± 18.94	90.43 ± 19.64	24.000	0.694
fNIRS				
TMT-B	2.53 ± 1.12	2.75 ± 2.16	24.000	0.694
Stroop Test	2.62 ± 1.58	4.37 ± 4.33	21.000	0.463

TMT-B: Trail Making Test Part B.

**Table 3 brainsci-13-01573-t003:** Comparison of executive functions between both groups (post-intervention).

	Experimental Group (*n* = 8)	Control Group (*n* = 7)	U	*p*
TMT-B (s)	15.64 ± 4.87	15.73 ± 2.56	24.000	0.643
Stroop Test (s)	77.62 ± 12.47	81.31 ± 16.32	24.000	0.643
fNIRS				
TMT-B	2.30 ± 0.99	2.33 ± 1.76	28.000	0.908
Stroop Test	2.40 ± 1.25	2.47 ± 1.96	25.000	0.728

TMT-B: Trail Making Test Part B.

**Table 4 brainsci-13-01573-t004:** Comparison of executive functions within each group (pre and post-intervention).

	Experimental Group (*n* = 8)				Control Group (*n* = 7)			
	Pre-Test	Post-Test	Z	*p*	Pre-Test	Post-Test	Z	*p*
TMT-B (s)	19.01 ± 5.25	15.64 ± 4.87	−2.380	0.017 *	17.79 ± 6.75	15.73 ± 2.56	−0.676	0.499
Stroop Test (s)	87.86 ± 18.94	77.62 ± 12.47	−2.380	0.017 *	90.43 ± 19.64	81.31 ± 16.32	−2.197	0.028 *
fNIRS								
TMT-B	2.53 ± 1.12	2.30 ± 0.99	−0.700	0.484	2.75 ± 2.16	2.33 ± 1.76	−0.507	0.612
Stroop Test	2.62 ± 1.58	2.40 ± 1.25	−0.280	0.779	4.37 ± 4.33	2.47 ± 1.96	−1.352	0.176

* *p* < 0.05, TMT-B: Trail Making Test Part B.

## Data Availability

The group data presented in this study are available upon request from the corresponding author. The individual data are not publicly available due to privacy and confidentiality.

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
