# Peer review of "Feasibility of Virtual Shopping Budget-Management Training on Executive Functions in Healthy Young Adults: A Pilot Study"

_brainsci, 2023, doi:10.3390/brainsci13111573_

Round 1
Reviewer 1 Report
Comments and Suggestions for Authors
The work is quite interesting for the increase in ecological validity it provides and taken the putative engagement of several executive functions on the intervention program. Nevertheless, I have serious concerns about the statistical methods chosen. Please see bellow my other comments.
1. Non-parametric statistics should be used given the small size of the samples. Even if tests of normality were run (not reported..) It should be preferable for such a small sample and especially given the null results reported.
2. Regarding the statistics, if the two-way interaction fails to report significant differences, then subsequent post-hoc test should not be run.
3. Regarding the introduction: i) the relation between daily living activities and executive functions should be better explained specially given it is such an important rationale for you study. ii) the role of the dlPFC and the functions attribute to it should be mention in the introduction section. A small paragraph might suffice.
4. In the participants section it should be mention the information that is only found afterwards, i.e, the number of participants and the existence of a control group and how they were assigned.
5. In the Measurement sections: i) In the “primary outcome” it is mention the dlPFc activation.. it is confusing since you mention it in the “secondary” ones. ii) It should be briefly mention what is the TMT and Stroop test putatively measuring. ii) No where in the manuscript it is explain the control group sessions. Did they have sessions? Comprised of what?
Minor
# “16” at the beginning of a sentence should read “sixteen”
# Throughout the manuscript: “executive function” should read “executive functions”
Author Response
Thank you for your constructive comments. We have revised our manuscript to reflect your comments as much as possible. Please refer to the attached response file.
#1. Non-parametric statistics should be used given the small size of the samples. Even if tests of normality were run (not reported..) It should be preferable for such a small sample and especially given the null results reported.
: Thank you for your feedback. We have re-analyzed our data using the Mann-Whitney U test for intergroup comparisons and the Wilcoxon signed-rank test for intragroup comparisons. As a result, we have found similar results to those from the previous parametric statistics.
Regarding the mention of not reporting the normality test, we apologize for any confusion. We did, in fact, conduct a normality test using the chi-square test, and the results are presented in Table 1. This information is important for transparency in our research and will contribute to providing a clear understanding to both the reviewers and readers.
#2. Regarding the statistics, if the two-way interaction fails to report significant differences, then subsequent post-hoc test should not be run.
: Your understanding is correct. It is a statistical convention not to conduct post-hoc tests when the interaction between two factors is not statistically significant. This principle reflects the idea that when there is no observed interaction between the two factors, conducting additional comparisons is not necessary, and it helps control the error rate while maintaining the reliability of research results.
In our analysis, we also confirmed the non-significance of the interaction between the two factors, and therefore, it is appropriate not to proceed with post-hoc tests. Additionally, by changing the statistical methods in our study to Mann-Whitney U test and Wilcoxon signed-rank test, we have ensured that post-hoc tests are not required.
We appreciate your feedback and your insights, and we are committed to making these adjustments to our manuscript.
#3. Regarding the introduction: i) the relation between daily living activities and executive functions should be better explained specially given it is such an important rationale for you study. ii) the role of the dlPFC and the functions attribute to it should be mention in the introduction section. A small paragraph might suffice.
: Thank you for your valuable feedback. We have strengthened the introduction section of the manuscript as you suggested.
Firstly, we have clarified the explanation of the relationship between daily life activities and executive functions. In our research, this relationship serves as one of the key foundations, and we have presented a more detailed description to ensure that readers can clearly understand the significance of our study.
We have added this information in the Introduction section (“Additionally, instrumental activities of daily living (IADL) encompass a range of activities crucial for independent living within the community. The absence of appropriate executive control may result in a higher likelihood of encountering difficulties in performing IADL effectively.”)
Additionally, we have included an explanation of the role and functions of the dlPFC in the introduction. The dlPFC plays a critical role in various cognitive functions and decision-making, which underscores its importance in our study. This information will help to convey the scientific basis and significance of our research more effectively.
We have added this information in the Introduction section (“One such brain area associated with executive functions is the dorsolateral prefrontal cortex (dlPFC). The dlPFC is conventionally acknowledged to play a crucial role in executive functions, which are high-level cognitive functions that support flexible goal-directed behaviors.”)
#4. In the participants section it should be mention the information that is only found afterwards, i.e, the number of participants and the existence of a control group and how they were assigned.
: Thank you for your feedback. We have made sure to add the design section to include the missing information. Specifically, we have explicitly indicated the number of participants and the presence of control groups, along with providing details on how they were assigned. This information plays a crucial role in understanding the research design and its implications for the results. It could enhance the readability and transparency of our paper in the future.
We have added this information in the Method section (“This study was a pilot study, and all participants were randomly assigned, with 8 participants in each, to either the experimental group or the wait-list control group using the Python computer language. The intervention consisted of 16 training sessions conducted twice a week over 8 weeks.”.
#5. In the Measurement sections: i) In the “primary outcome” it is mention the dlPFC activation. It is confusing since you mention it in the “secondary” ones. ii) It should be briefly mention what is the TMT and Stroop test putatively measuring. ii) No where in the manuscript it is explain the control group sessions. Did they have sessions? Comprised of what?
: Thank you for your thoughtful suggestions. We take your feedback into account and have made the following revisions to the measurement section:
- i) Considering the potential confusion regarding the mention of dlPFC activation in the "Primary outcome" section, we have removed the reference to dlPFC activation from this section. This information has been presented in the "Secondary outcome" section instead.
- ii) We have included brief explanations of the TMT and Stroop tests to clarify the content they assess. These explanations would aid readers in understanding the significance of these tests.
We have added this information in the Method section (TMT-B: “TMT-B instructs participants to connect numbers and letters alternately in a consecutive sequence. This assessment is related to executive function and evaluates cognitive flexibility.”; Stroop test: “This assessment is a tool for evaluating cognitive flexibility and working memory. There were four colors: red, yellow, blue, and black. Participants were requested to read a total of 112 words as quickly as possible.”)
iii) We appreciate your observation regarding the control group. The control group in our study indeed consisted of a wait-list control group, and they did not receive any intervention sessions. They were primarily utilized for pre- and post-assessments. We acknowledge that this information was not explicitly stated in the manuscript, and we appreciate your attention to detail.
We have added this information in the Method section (“This study was a pilot study, and all participants were randomly assigned, with 8 participants in each, to either the experimental group or the wait-list control group using the Python computer language.”)
Minor
# “16” at the beginning of a sentence should read “sixteen”
: We have revised it.
# Throughout the manuscript: “executive function” should read “executive functions”
: We have revised it.

Reviewer 2 Report
Comments and Suggestions for Authors
The paper presents an empirical study investigating virtual shopping budget-management training and executive functions in healthy young adults.
The research topic is of interest for cognitive research.
The manuscript is well written and understandable.
Important strengths include the novel approach and the detailed assessments.
However, the sample is very small. Only 16 participants (divided into 2 experimental groups). I learned that this is a pilot study. Nevertheless, I am wondering if estimates are reliable. Has an a-priori power analysis been conducted? Why publishing the pilot study and not waiting for the real study with a bigger sample? At least, these important issues await a more detailed explication.
Only non-significant results for the main analyses. How does this study help to advance research in detail?
For these reasons, the title (speaking about effects) is misleading, because there were no effects observed.
Only TMT and Stroop have been assessed as cognitive outcomes. A broader set of executive functioning tests are needed.
The practical implications need to be presented in more detail.
Author Response
Thank you for your constructive comments. We have revised our manuscript to reflect your comments as much as possible. Please refer to the attached response file.
#1. The sample is very small. Only 16 participants (divided into 2 experimental groups). I learned that this is a pilot study. Nevertheless, I am wondering if estimates are reliable. Has an a-priori power analysis been conducted? Why publishing the pilot study and not waiting for the real study with a bigger sample? At least, these important issues await a more detailed explication.
: We appreciate your thoughtful comments and concerns and are grateful for the time you've dedicated to reviewing our work. Your observations regarding the small sample size are indeed valid. In fact, this study was a pilot study, with its primary purpose being the collection of initial data and the validation of the research design. Your feedback has been particularly helpful, and in response to this, we have explicitly indicated the pilot nature of this study by adding a dedicated section, preceding the participants section.
Furthermore, you have correctly pointed out the absence of an a priori power analysis. This was due to the limited availability of prior research in the area. However, we do understand the importance of power calculations in determining sample size and assure you that we'll address this concern when designing and conducting future studies in this domain.
#2. Only non-significant results for the main analyses. How does this study help to advance research in detail?
: We appreciate your question regarding the significance of our study despite finding only non-significant results for the main analyses. While our primary outcomes did not show statistically significant effects, this study plays a crucial role in advancing research by focusing on the importance of more comprehensive measurement methods when assessing the impact of cognitive training.
It's important to note that our research goes beyond the mere observation of statistical significance. In fact, it underscores the need for more refined and sophisticated measurement techniques to capture nuanced changes in cognitive functions. Our study recognizes that traditional assessments may not provide the whole picture, especially when dealing with factors such as mood states and individual variability, which can influence participants' performance (Chepenik, L. G.; Cornew, L. A.; Farah, M. J. The influence of sad mood on cognition. Emotion. 2007, 7(4), 802). As highlighted in previous studies, utilizing neurofeedback based on brain signals, measured using techniques such as fNIRS, allows for the personalization of training difficulty levels (Jeun, Y. J.; Nam, Y.; Lee, S. A.; Park, J. H. Effects of Personalized Cognitive Training with the Machine Learning Algorithm on Neural Efficiency in Healthy Younger Adults. International Journal of Environmental Research and Public Health. 2022, 19(20), 13044).
In essence, our research acts as a catalyst for encouraging the field to move towards more advanced assessment tools, thus expanding the understanding of cognitive enhancements. These tools have the potential to reveal subtle changes that traditional assessments might overlook. This personalization and gradation of cognitive training, based on continuous monitoring of participants' brain activity, are essential for more effective and tailored interventions.
We have added this information in the Discussion section (“This study emphasizes the importance of advanced measurement techniques, such as neuroimaging, in evaluating cognitive training. Beyond statistical significance, it is essential to measure objective evidence, such as neural efficiency, to confirm intervention effects, surpassing the limitations of traditional assessments influenced by external factors like mood states.”)
#3. For these reasons, the title (speaking about effects) is misleading, because there were no effects observed.
: Thank you for your feedback on the title, and we appreciate your understanding of our perspective. While the main analysis did not yield significant effects, the primary aim of this study was indeed to explore specific effects. Therefore, we were taking your point into consideration and have made revisions to the title, acknowledging the importance of reflecting the absence of observed effects while aligning it with the content and purpose of the paper.
We have decided to change the title from 'effects' to 'feasibility,' which will distinctly reflect the lack of observed results in the main findings and better match the paper's content and objectives. This modification aims to enhance the clarity of the paper and convey the research content more accurately.
#4. Only TMT and Stroop have been assessed as cognitive outcomes. A broader set of executive functioning tests are needed.
: We would like to express our gratitude for the valuable feedback provided by the peer reviewer. While our study has focused on the TMT and Stroop tests as cognitive outcomes, we acknowledge the importance of including a broader range of executive function tests to enhance the comprehensiveness of our research findings.
Furthermore, we also recognize the significance of addressing the limitations of our current study. Therefore, we will make it a point to clearly outline these limitations and discuss our efforts to overcome them in future research.
We have added this information in the Discussion section (“Fourthly, despite the presence of various sub-elements in executive functions, this study only measured some aspects using the TMT-B and Stroop test. Future research should utilize a variety of executive function assessment tools for additional analysis, particularly considering a larger sample with executive function impairments.”)
Your insights and recommendations are highly appreciated, and we are committed to utilizing them to improve and advance our research. Please feel free to share any further suggestions or questions as your feedback is invaluable to us.
#5. The practical implications need to be presented in more detail.
: Thank you for your feedback. We appreciate your interest in the non-significant results observed in the main analysis. Your question about how these results may contribute from a clinical perspective is indeed important.
Due to our small sample size, our study did not achieve statistical significance. However, it is essential to recognize that such results can still be valuable in the fields of clinical research and healthcare. For instance, our research can contribute to the development of training programs for effective virtual shopping budget management. Thus, our study can act as a foundational step in the creation of practical virtual shopping budget-management training.
Moreover, these research findings can be applied to rehabilitation and elderly care programs for the elderly or individuals with brain injuries. Training in a virtual environment can help enhance cognitive abilities and improve daily functioning by simulating real-life situations (Bauer, A. C. M.; Andringa, G. The potential of immersive virtual reality for cognitive training in elderly. Gerontology. 2020, 66(6), 614-623). Considering these practical implications, our research results emphasize their potential utility in various fields, with the capacity to enhance both individual and societal well-being.
We have added this information in the Discussion section (“Additionally, despite the small sample size and nonsignificant results in primary analyses, our research can contribute to the development of effective virtual reality education programs for shopping budget management in the future. Hence, our study can serve as a foundational step towards creating practical virtual reality education for budget management during shopping. Furthermore, these findings can also benefit rehabilitation and elderly care programs. Training in a virtual environment enhances cognitive abilities and daily functioning, with potential for broader societal impact.”)
